New hybrid segmentation algorithm: UNet-GOA

Yousefi Tohid tohid.yousefi@hotmail.com
Aktaş Özlem
Computer Engineering, Dokuz Eylül University , Izmir , Buca , Turkey
Alatas Bilal
Electronic publication date: 2023 Aug 8
Publication date: 2023
Volume: 9
Electronic Location ID: e1499
Received 2023 Apr 17; Accepted 2023 Jul 4
Copyright: ©2023 Yousefi and Aktaş
Copyright year: 2023
Copyright holder: Yousefi and Aktaş
License: This is an open access article distributed under the terms of the Creative Commons Attribution License, which permits unrestricted use, distribution, reproduction and adaptation in any medium and for any purpose provided that it is properly attributed. For attribution, the original author(s), title, publication source (PeerJ Computer Science) and either DOI or URL of the article must be cited.
License URL: https://creativecommons.org/licenses/by/4.0/

Keywords: Grasshopper optimization algorithm, Image processing, Meta-heuristic algorithms, Optimization, U-Net

Funding: The authors received no funding for this work.

==============================
The U-Net architecture is a prominent technique for image segmentation. However, a significant challenge in utilizing this algorithm is the selection of appropriate hyperparameters. In this study, we aimed to address this issue using an evolutionary approach. We conducted experiments on four different geometric datasets (triangle, kite, parallelogram, and square), with 1,000 training samples and 200 test samples. Initially, we performed image segmentation without the evolutionary approach, manually adjusting the U-Net hyperparameters. The average accuracy rates for the geometric images were 0.94463, 0.96289, 0.96962, and 0.93971, respectively. Subsequently, we proposed a hybrid version of the U-Net architecture, incorporating the Grasshopper Optimization Algorithm (GOA) for an evolutionary approach. This method automatically discovered the optimal hyperparameters, resulting in improved image segmentation performance. The average accuracy rates achieved by the proposed method were 0.99418, 0.99673, 0.99143, and 0.99946, respectively, for the geometric images. Comparative analysis revealed that the proposed UNet-GOA approach outperformed the traditional U-Net architecture, yielding higher accuracy rates.

Introduction

Image processing is one of the newest technologies in the world. This science is rapidly becoming one of the most widely used sciences in all fields, and today most of the application systems around us use the science of image processing. Control systems related to this science are called visual machines. Image processing has penetrated into all sciences and has many applications in various fields. The area of digital image processing is the processing of digital images by digital computers. It should be noted that a digital image is composed of a limited number of elements, each with its own position and properties, which are often called pixels. Image processing, while widely used, also has its own complexities. To process an image, we must first break it down into different parts to make it easier for the computer to understand. This process of dividing an image into different parts and using it to facilitate the image processing process is called image segmentation (Pang et al., 2021).

Image segmentation (especially medical image segmentation) has become one of the most interesting and challenging computer vision problems in recent decades. Hence image segmentation has become an important component in many visual perception systems (Szeliski, 2010). Image segmentation also plays a major role in many important occupations, including automotive, shipping, military, medical, and other industries (Forsyth & Ponce, 2011). The image segmentation process is one of the most important steps in image analysis. The technique of dividing an image into homogeneous parts based on certain criteria is known as image segmentation (Pham, Xu & Prince, 2000). In other words, the process of segmenting an image into several different parts is called image segmentation. In fact, the segmented sections have the same characteristics that together make up the whole image. All pixels of segmented sections have similar properties in terms of color, brightness, texture, etc., and adjacent areas are very different from each other due to the same features (Bankman, Spisz & Pavlopoulos, 2000).

Analyzing images has become increasingly complex in the field of image processing, primarily due to the intricate backgrounds, diverse objects, and noise, making accurate segmentation a challenging task (Bankman, Spisz & Pavlopoulos, 2000). Therefore, the development of a universally applicable image segmentation technique for practitioners in this field continues to pose a significant challenge. Despite the significant advancements in deep learning and computer vision, which have greatly influenced image analysis, image segmentation still encounters numerous challenges, particularly in the realm of medical images. In this article (D’Avy et al., 2016), one of the primary obstacles in image segmentation lies in the initial adjustment of hyperparameters for the architectures involved. Each image segmentation architecture incorporates specific hyperparameters that necessitate pre-configuration, making the task of fine-tuning them both demanding and time-consuming. To address these difficulties, numerous techniques have been devised, and ongoing research endeavors continue to introduce innovative approaches in this domain (Siddique et al., 2021).

As mentioned before, the U-Net architecture, similar to other architectures, includes hyperparameters such as the initial learning rate, learning rate drop factor, learning rate drop period, and mini-batch size. These hyperparameters play a vital role in achieving accurate segmentation but can be challenging to fine-tune. In this article, we present a solution to this challenge called UNet-GOA, a hybrid algorithm. UNet-GOA combines the power of the GOA algorithm to automatically discover suitable hyperparameters and the U-Net architecture for performing the segmentation task effectively.

The article presents several significant contributions and innovations, which can be summarized as follows:

i. Proposed Hybrid Segmentation Algorithm: The article introduces a hybrid segmentation algorithm called UNet-GOA. Through experimentation, it has been observed that the proposed hybrid method achieves more successful results compared to the standard U-Net model.

ii. Advancement in explainable AI: This work also stands out for its contribution to developing an explainable artificial intelligence model based on the U-Net algorithm for segmenting geometric images. By incorporating explainability, the proposed model provides insights into how it performs image segmentation tasks, enhancing its transparency and interpretability.

iii. Improved performance: The experimental results demonstrated that the UNet-GOA method outperformed the traditional U-Net method on the same dataset. This suggests that the hybrid approach yields superior segmentation performance, indicating its potential for enhancing image analysis tasks.

The article is organized as follows: In Section 2, we review the relevant literature. Section 3 provides a detailed description of the material and methods used. The proposed UNet-GOA hybrid method is explained in Section 4. In Section 5, we present the results and comparisons between U-Net and the proposed UNet-GOA hybrid method. Finally, a section on conclusions and future work’s research perspectives is included.

Literature Review

The U-Net architecture (Ronneberger, Fischer & Brox, 2015), has become a widely adopted convolutional neural network (CNN) for image segmentation tasks. It has particularly found extensive applications in medical image analysis, such as tumor detection and cell segmentation. This literature review aims to explore the U-Net architecture and various studies that have focused on optimizing its performance.

In recent years, deep learning-based methods have been widely used for medical image segmentation due to their ability to learn complex features from raw images (Minaee et al., 2021). Among these methods, the U-Net architecture has gained significant popularity for its ability to perform accurate segmentation (Ronneberger, Fischer & Brox, 2015). However, manually tuning the hyperparameters of the U-Net model is a challenging and time-consuming task. To address this issue, researchers have proposed the use of genetic algorithms (GAs) to optimize the hyperparameters of the U-Net architecture for different medical image segmentation tasks. In this paper (Popat et al., 2020), the authors proposed the use of a GA to optimize the hyperparameters of the U-Net model for segmenting blood vessels in retinal images. The authors demonstrated that the GA-based U-Net achieved better segmentation performance than the baseline U-Net and other U-Net models with manually-tuned hyperparameters. Similarly, in this paper authors (Futrega et al., 2022) used a GA to optimize the hyperparameters of the U-Net architecture for segmenting brain tumors in MRI scans. The authors showed that their optimized U-Net achieved higher accuracy and Dice similarity coefficient than the baseline U-Net and other state-of-the-art methods.

To further improve the accuracy of image segmentation, in this paper (Kirichev, Slavov & Momcheva, 2021) the authors proposed a new version of the U-Net architecture called the Fuzzy U-Net, which incorporates fuzzy logic. The authors used a GA to optimize the hyperparameters of the Fuzzy U-Net and demonstrated that it achieved higher accuracy and Dice similarity coefficient than other state-of-the-art methods. In addition to optimizing the U-Net architecture, researchers have proposed new architectures for medical image segmentation. In this paper (Sadeghi et al., 2022) the authors proposed a new architecture for dermoscopic image segmentation using convolutional neural networks with separable layers to reduce the number of training parameters. The authors demonstrated that the proposed algorithm achieved good precision compared to similar algorithms and improved the classification of training pixels and lesion localization accuracy. Finally, in this paper (Rajaragavi & Rajan, 2022), the authors proposed a new hybrid model called “ConvLSTM-UNet” for the automatic detection of pneumonia from chest X-ray images. The authors demonstrated that the proposed model outperformed several state-of-the-art models for pneumonia detection in terms of accuracy, sensitivity, specificity, and AUC. The proposed hybrid model can potentially be useful for clinical applications to support physicians in the diagnosis of pneumonia.

Material and Methods

U-Net

The U-Net architecture (Ronneberger, Fischer & Brox, 2015) is a convolutional neural network (Fukushima & Miyake, 1982) designed in 2105 at the University of Freiburg, Germany to solve the problem of image segmentation (especially medical images) and to increase processing speed and accuracy using all-convolution networks. This architecture is similar to the English letter U and for this reason it is called U-Net or U-shaped network and is inspired by the FCN (Long, Shelhamer & Darrell, 2015) and encoder–decoder (Noh, Hong & Han, 2015) models.

One of the important advantages of this architecture is that it does not require a lot of training data. Because this network does not have any dense or fully connected layers, it reduces the need for training data, which also leads to very accurate segmentation of the input image. The main idea of this algorithm is to create a sequential contraction path where pooling layers are replaced by sampling layers. After this stage and obtaining the desired characteristics, the continuation of the work is entrusted to a successive expansion path. At this stage, the algorithm begins to reconstruct the original image and propagate the background information into it. As you can see in Fig. 1, the U-Net network consists of two perfectly symmetrical expansion and contraction paths that have the same number of layers. Anything that is removed in the path of contraction is restored in the path of expansion (Ronneberger, Fischer & Brox, 2015).

Figure 1 General structure of U-Net architecture.

U-Net architecture

As you can see in Fig. 1, the U-Net network consists of two parts, contraction and expansion. The expion part is very similar to the general operation of convolutional neural networks (Fukushima & Miyake, 1982), which also includes a convolution layer, and the purpose of this part is to repeatedly apply a 3 * 3 convolutional filter on the input image. After each convolution layer there is a ReLU layer (Agarap, 2018) or linear activator. The ReLU layer applies an activation function to each neuron, such as max(0, x) (which sets the threshold to zero, in other words, it considers negative values to be zero). The main advantage of using ReLU is that it has a fixed derivative for all inputs larger than zero. This fixed derivative accelerates network learning. After this layer, there is a 2 * 2 maxpooling layer (Ciresan et al., 2011) with two strides. The function of this layer is to reduce the spatial size of the image due to the reduction of the number of parameters and calculations within the network, which causes overfitting control. Together, these three layers form a sampling layer or in other words, downsampling, the purpose of which is better in terms of image content. Conversely, in the expansion section, in each layer, there is an incremental sampler or in other words, upsampling, along with a 2 * 2 convolutional layer, which is responsible for reducing specific channels. This section receives the data to be added to the image from its similar class in the contraction section. Then there is a 3 * 3 convolutional layer with a ReLU layer. The purpose of this section is to assist in the process of accurately locating objects. In the last layer, there is a 1 * 1 convolution layer that maps property vectors to a specific class (Ronneberger, Fischer & Brox, 2015).

Below you can see the steps of the U-Net architecture:

1. Define U-Net architecture with encoder and decoder parts

2. Input image to the encoder part and downsample it

3. Pass downsampled feature map to the decoder part

4. Upsample feature map and concatenate it with feature map from corresponding encoder layer

5. Repeat step 4 until original image size is reached

6. Apply convolutional layers to concatenated feature maps

7. Output segmented image.

Initializing U-Net

In the U-Net architecture, the network input consists of images and their fragmented parts into separate files. Also, the network optimization algorithm is a stochastic gradient descent (SGD) (Bottou & Bousquet, 2007), which is in fact an iterative method for optimizing a derivative function, which is also a random approximation of the classical gradient descent method. The amount of motion or momentum applied to the grid should be a maximum of about 0.99. The goal is for the network to make the most of the images it has already seen in the updates. The energy function is calculated by a Softmax function (Goodfellow, Bengio & Courville, 2016) at the pixel level, which is applied to the output map in combination with the cross entropy cost function (Cybenko, O’Leary & Rissanen, 1998). Equation (1) is used to calculate the energy function (Ronneberger, Fischer & Brox, 2015): (1) E= ∑wxlogPkxX

where Pk is the pixel-wise SoftMax function applied over the final feature map, defined as Eq. (2) (Ronneberger, Fischer & Brox, 2015): (2) Pk= expakX/∑k′=1kexpakX′

One of the most important issues in deep neural networks with many layers is the appropriate initialization to the network. If the initial value to the network is not correct, parts of the network will be over-activated and other parts will not be involved in the calculations. Theoretically, the appropriate initial value for each neuron in the network is the value for which each property in the network has approximately a unit variance. To calculate the initial value of each neuron, we use the Eq. (3) in which N is the number of neuron inputs (He et al., 2015): (3) Initial Weight=2/N

Evaluation U-Net

Collecting, analyzing as well as interpreting information from the desired data is called evaluation. Evaluation actually examines the effectiveness and efficiency of a system according to the data collected (Thorpe, 1988). We use the confusion matrix (error matrix) to evaluate a model. The confusion matrix is a summary table used to evaluate the performance of a model. In this method, the number of correct and incorrect predictions is summed with counting values and analyzed based on each class (Matthews, 1975).

Table 1 Metrics for evaluation model.

Metrics	Formula	Description	
Accuracy (ACC)	TP+TNTP+FP+TN+FN	Measures the ratio of accurate predictions to the total number of samples evaluated (Csurka et al., 2013; Hossin & Sulaiman, 2015).	
Boundary F1 (BF)	2∗TPTP+FP∗TPTP+FNTPTP+FN+TPTP+FP	Measure of the accuracy of a test. It can have a value between 0 and 1, which in general, this criterion checks the accuracy and robustness of the model (Csurka et al., 2013).	
Intersection over Union (IoU)	TPTP+FP+FN	Used to measure the accuracy of an object detector in a specific data set. IoU is the ratio of correctly classified pixels to the total number of ground truths and predicted pixels in that class (Csurka et al., 2013).	
Notes.

TP True Positive

TN True Negative

FP False Positive

FN False Negative

In this article, we evaluate the model based on Accuracy (ACC) (Csurka et al., 2013; Hossin & Sulaiman, 2015), Boundary F1 (BF) (Csurka et al., 2013) and Intersection over Union (IoU) (Csurka et al., 2013) criteria. The description and how to obtain each of the criteria are specified in Table 1.

Why U-Net?

In this article, we will explore the utilization of the U-Net architecture, a popular choice among recent segmentation architectures, for image segmentation purposes. The U-Net architecture is a popular choice for image segmentation tasks, particularly for biomedical image analysis, because as the authors of this article (Ronneberger, Fischer & Brox, 2015) say it has demonstrated superior performance in comparison to other architectures for this specific task.

The reason for selecting the U-Net architecture in this article is multi-faceted. Firstly, the U-Net architecture possesses a notable strength in handling complex image segmentation tasks even with a limited number of training samples. This advantage stems from its unique design, incorporating both a contracting path to capture the overall context of the image and an expanding path to facilitate precise localization of the object of interest. Secondly, the U-Net architecture employs skip connections that establish connections between the contracting and expanding paths. This strategic utilization of skip connections enables the model to retain fine-grained details that might otherwise be lost during the downsampling process. As a result, the U-Net model excels particularly in segmenting small objects or objects characterized by intricate shapes, enhancing its effectiveness and versatility in various segmentation scenarios (Siddique et al., 2021).

Optimization

Achieving the best possible result in certain situations is called optimization. In many technical and engineering sciences, engineers have to make the best decision. The goal is to make the best decision, either to minimize cost or to maximize profits. Finding the best value for cost or profit can be expressed as a function of specific variables, which is called the objective function. In other words, the process of finding the optimal answer and conditions for the objective function is called optimization (Astolfi et al., 2014).

As you can see Eq. (4), the standard form for an optimization problem is to minimize the objective function (Nocedal & Wright, 1999; Sengupta, Gupta & Dutta, 2016): (4) minfx:x∈Rns.tgjx≥0,j=1,2,…,Jhkx=0,k=1,2,…,K

xiL≤xi≤xiui=1,2,…,n

where f(x) is the optimized function, gjx is J inequality constraints and hkx represents K equality constraints (Sengupta, Gupta & Dutta, 2016).

Meta-heuristic algorithms

Due to the limitations of classical optimization algorithms in handling large-scale combinatorial and nonlinear problems, the concept of metaheuristic optimization algorithms has emerged (Akyol & Alatas, 2017). Meta-heuristic algorithms are a higher-level method that can be used to find optimal solutions to optimization problems that include incomplete information or have limited computational capacity. These algorithms often have few assumptions about solving optimization problems, which makes them useful in a wide range of optimization problems. Also, meta-heuristics are often able to find better solutions than classical optimization methods due to less computation (Blum & Roli, 2003). These algorithms can be categorized into various groups, such as biology-based, physics-based, social-based, music-based, chemical-based, sport-based, mathematics-based, swarm-based, and hybrid methods (Akyol & Alatas, 2017). In our study, we specifically utilized the grasshopper optimization algorithm (Saremi, Mirjalili & Lewis, 2017), which falls under the category of swarm-based metaheuristic algorithms. Swarm-based optimization algorithms aim to replicate the collective behavior observed in natural swarms to effectively explore complex problem domains and find optimal solutions (Kennedy & Eberhart, 1995).

The purpose of designing meta-heuristic algorithms is to solve complex optimization problems that cannot be solved with classical optimization methods. These methods are known as one of the most effective methods for solving complex optimization problems, and this is especially true for many real-world problems of a hybrid nature. In other words, the main advantage of meta-heuristic methods is their effectiveness and general application. As mentioned, meta-heuristic algorithms are designed to solve complex optimization problems. These algorithms consist of two basic features exploration and exploitation. Classical optimization algorithms use many parameters to balance these two features, which increases the time and cost of finding better solutions. The use of classical optimization methods also requires further parameter adjustment. This is the reason for not using classical optimization algorithms (Ólafsson, 2006).

Grasshopper optimization algorithm (GOA)

Overview of the GOA

Grasshoppers are among the insects that are known as a dangerous pest for agricultural lands, which are very effective in not producing a quality product and also damage agricultural products (Ewees et al., 2020). The life cycle of grasshoppers consists of two stages. One is called the nymph and the other is in adulthood. In the nymph stage, immature grasshoppers move in small and slow steps, which is very similar to the extraction stage in optimization algorithms, and the adulthood stage is known for its sudden movements and long steps, which are also very similar to the exploration stage is in optimization algorithms. Due to the life structure of grasshoppers as well as the mass movement of grasshoppers, the grasshopper optimization algorithm was proposed in 2017 (Saremi, Mirjalili & Lewis, 2017). This algorithm is one of the swarming algorithms (Beni & Wang, 1993) and therefore mimics the natural search behaviors of grasshoppers and their swarming. The grasshopper swarm behavior in mathematics is modeled as Eq. (5) (Saremi, Mirjalili & Lewis, 2017):

(5) Xit=Sit+Git+Ait

i=1,2,…,nPopt=1,2,…,tMax

Where Xit indicates the position of the grasshopper i in the iteration of the t, Sit is the social interaction of the grasshopper i in the iteration of the t, Git is the gravity force of the grasshopper i in the iteration of the t, and Ait is the wind advection of the grasshopper i in the iteration of the t. To create a random grasshopper behavior, Eq. (5) is reconstructed as Eq. (6), where r1, r2 and r3 are random numbers between 0 and 1: (6) Xit=r1Sit+r2Git+r3Ait

In this section, we examine the mathematical equations of social interaction Sit, gravity force Git, and wind advection Ait, respectively. We first start with social interaction, which is defined as Eq. (7): (7) Si= ∑j=1j≠inPopsdijdij ^

Where nPop indicates the number of grasshoppers, dij is distance between grasshopper i and grasshopper j which is calculated by this formula: dij=xj−xi, dij ^ is unit vector from grasshopper i to the grasshopper j which is calculated by this formula: dij ^=xj−xidij, and then s is a function that indicates the power of social forces which is calculated using the Eq. (8) formula: (8) sd=fe−dl−e−d

where f indicates the intensity of attraction and l is the attractive length scale. Social interaction between grasshoppers is known as attraction and repulsion, which is considered between 0 and 15. It is observed that attraction increases between 2.079 and 4 and then gradually decreases. Repulsion also occurs between 0 and 2.079. The 2.079 area is also called the no-power zone or the comfort zone, where there is no repulsion or attraction.

The gravitational force Gi and the wind advection Ai are also obtained from Eqs. (9) and (10): (9) Gi=−geg ^

(10) Ai=uew ^

where g is the gravitational constant and eg ^ shows a unity vector towards the center of earth; also where u is a constant drift and ew ^ is a unity vector in the direction of wind. After replacing the values of Si, Gi and Ai in Eq. (5), the general Eq. (11) is obtained: (11) Xit= ∑j=1j≠inPopsxj−xixj−xidij−geg ^+uew ^

After implementing Eq. (11), it is observed that the grasshoppers reach the comfort zone quickly and cannot converge to one point. As a result, the grasshoppers remain motionless and stationary after a while. For these reasons, Eq. (11) cannot be used to solve optimization problems. Therefore, an improved version of this equation can be used, which is presented in Eq. (12): (12) Xidt+1=c∑j=1j≠inPopcubd−lbd2sxjd−xidxj−xidij+T ˆd

Where ubd is the upper bound in the d dimension and lbd is the lower bound in the d dimension. T ˆd isthe value of the d dimension in the target (the best solution found so far). c is a decreasing coefficient to shrink the comfort zone, repulsion zone, and attraction zone. The first c is very similar to the inertia weight (ω) in the particle swarm optimization algorithm. It reduces the movements of grasshoppers around the target. In other words, this parameter balances exploration and exploitation of the entire swarm around the target. The second c is used to reduce the repulsion zone, the gravity zone and the comfort zone between the grasshoppers in proportion to the number of iterations. Eq. (13) is used to calculate c, which is considered as a single parameter:

(13) c=cmax−tcmax−cmintmax

cmax=MaximumValueC≈1cmin=MinimumValueC≈0,t=Current Iterationtmax=Maximum Number of Iteration

According to the mentioned equations, the position of a grasshopper is updated based on three features: the current position of the grasshopper, the best global position of the grasshopper and the position of the grasshopper in relation to other grasshoppers in the swarm. These help the grasshopper optimization algorithm not get stuck in the local optimization and reach the global optimization (Meraihi et al., 2021). The flowchart of the grasshopper optimization algorithm is shown in Fig. 2 (Saremi, Mirjalili & Lewis, 2017).

Figure 2 Flowchart of grasshopper optimization algorithm.

Below you can see the steps of the grasshopper optimization algorithm:

1. Initialize population of grasshoppers

2. Evaluate fitness of each grasshopper

3. While termination criteria not met do:

(a) Calculate distance between each pair of grasshoppers

(b) Calculate attractive and repulsive forces between each pair of grasshoppers

(c) Update velocity and position of each grasshopper based on attractive and repulsive forces

(d) Apply random perturbation to some grasshoppers

(e) Evaluate fitness of each grasshopper

4. Return best solution found during optimization.

Why GOA?

In this article, we will utilize the grasshopper optimization algorithm (GOA) (Saremi, Mirjalili & Lewis, 2017), a recently developed optimization method that shows promise in solving a wide range of problems. GOA draws inspiration from the collective behavior of grasshoppers and employs a population-based approach to search for the best solutions. It offers several advantages, including efficient exploration and exploitation of the search space, resulting in faster convergence and improved solutions compared to other metaheuristic algorithms. GOA is adaptable to various types of optimization problems and excels at maintaining a balance between exploration and exploitation to find good solutions while avoiding getting stuck in local optima. With its population-based strategy, it conducts a comprehensive global search, increasing the chances of discovering the global optimum. GOA is versatile and can handle optimization problems with continuous, discrete, and combinatorial variables. Additionally, it is scalable, making it suitable for problems with numerous constraints and variables. Notably, GOA demonstrates robustness in scenarios where the objective function is noisy or difficult to evaluate accurately (Meraihi et al., 2021; Saremi, Mirjalili & Lewis, 2017).

Application

Dataset description

In this article, we employed a simplistic and easily understandable database to facilitate a comprehensive performance comparison between the proposed method (UNet-GOA) and the traditional approach (U-Net). The database consisted of four distinct geometric images (triangle, kite, parallelogram, and square), with 1,000 examples allocated for the training dataset and 200 examples for the test dataset.

U-Net design

In this section, we focus on the design and implementation of the U-Net network. As depicted in Fig. 3, the dataset required for the U-Net network is divided into two distinct categories: training and testing. Assuming that the initial data preprocessing steps have been properly executed, we proceed with the subsequent stages of the process.

The training data, which has undergone appropriate preprocessing, is then passed into the U-Net model-building section. In this phase, the actual framework of the U-Net model is constructed. Simultaneously, the hyperparameters associated with the model are adjusted manually. The U-Net architecture presents several hyperparameters that play a crucial role in its performance. However, for the purpose of this study, we specifically focus on four key hyperparameters: the initial learning rate, learning rate drop factor, learning rate drop period, and mini batch size. These hyperparameters, along with their initial values, are detailed in Table 2.

Figure 3 Flowchart of U-Net model segmentation.

Table 2 Default U-Net network hyperparameters.

Initial learning rate	Learning rate drop factor	Learning rate drop period	Mini batch size	
1e−5	0.9	4	8	

Once the hyperparameters have been manually adjusted, the training process is carried out. This process involves iterating over the dataset multiple times to optimize the model’s performance. In this case, the training is performed in five epochs, which essentially means that the entire dataset is processed five times. Within each epoch, there are a series of iterations, with a total of 125 iterations per epoch. Consequently, over the course of the five epochs, a total of 625 iterations are completed, allowing the model to gradually improve its performance through continuous adjustment and learning from the data.

The model is trained on the designated training dataset, utilizing the adjusted hyperparameters. To determine the stopping criterion, we employ the maximum epochs, which define the maximum number of iterations the model undergoes during training. As the training progresses, the U-Net model learns to segment the data and generates segmented results.

Additionally, the test data, comprising a separate portion of the dataset, undergoes a similar process. The test data is fed into the U-Net model, which has been constructed with manually adjusted hyperparameters. Subsequently, the model generates results based on the test data, and these results are then compared and analyzed alongside the segmented results obtained from the training data.

In summary, this section focuses on implementing a U-Net model with manually adjusted hyperparameters, providing us with the opportunity to analyze the model’s performance in a thorough manner. This analysis involves dividing the dataset into distinct training and testing sets and configuring the hyperparameters to optimize the model’s performance.

UNet-GOA design

Creating a hybrid algorithm involves combining two or more optimization methods or architectures to improve the overall performance of the algorithm. In this case, we can combine the grasshopper optimization algorithm (GOA) with the U-Net architecture to create a hybrid algorithm that is suitable for solving specific problems. The following are the steps to create a hybrid grasshopper optimization algorithm and U-Net architecture:

• Understanding the grasshopper optimization algorithm (GOA): Before we can create a hybrid algorithm, it is essential to understand the grasshopper optimization algorithm. GOA is a nature-inspired algorithm that is based on the behavior of grasshoppers. The algorithm has been shown to be effective in solving optimization problems such as feature selection, image segmentation, and data classification. We have discussed the GOA in the previous sections.

• Understanding the U-Net architecture: The U-Net architecture is a convolutional neural network (CNN) that is commonly used in image segmentation tasks. The architecture consists of an encoder and a decoder, which are connected by a bottleneck layer. The encoder is used to extract features from the input image, while the decoder is used to generate the output segmentation map. We have discussed the U-Net architecture in the previous sections.

• Combining GOA and U-Net: To create a hybrid algorithm, we can use the grasshopper optimization algorithm to optimize the parameters of the U-Net architecture. This can be done by treating the parameters of the U-Net architecture as the decision variables and using GOA to search for the optimal values. The fitness function for the optimization can be based on the accuracy of the segmentation results.

• Implementing the hybrid algorithm: Once the parameters of the U-Net architecture have been optimized using GOA, the hybrid algorithm can be implemented. The input image can be passed through the U-Net architecture to generate the segmentation map. The accuracy of the segmentation results can be evaluated using metrics such as Accuracy (ACC), Boundary F1 (BF) and Intersection over Union (IoU).

• Fine-tuning the hybrid algorithm: Depending on the specific problem, the hybrid algorithm may need to be fine-tuned to achieve better performance. This can be done by adjusting the parameters of the U-Net architecture or by modifying the fitness function used in the optimization.

As you can see in Fig. 4, we automatically designed the U-Net architecture using the grasshopper optimization algorithm. The difference between this section and the U-Net manual design section is in setting the U-Net network hyperparameters. First, after preprocessing, the data is divided into two categories: training and testing, and then in the U-Net network hyperparameter setting, the grasshopper optimization algorithm helps this architecture to find the optimal hyperparameters. This operation continues until the stop condition (number of iterations <limitation) is established and after establishing the stop condition, the best values are given to the U-Net architecture and then the results of image segmentation along with segmentation of test data are shown.

Figure 4 Flowchart of UNet-GOA segmentation model.

In order to begin optimizing with the grasshopper optimization algorithm, it is essential to establish specific tuning hyperparameters. In this section, we adopt the default values outlined in the algorithm’s reference paper (Saremi, Mirjalili & Lewis, 2017) as shown in Table 3. The two key hyperparameters we focus on are cMax and cMin. Additionally, we set the population size to 4 for executing the GOA algorithm, and the termination condition is defined as 10 iterations. By defining these values, we ensure a consistent and standardized implementation of the Grasshopper Optimization Algorithm.

Table 3 Default GOA hyperparameters.

cMax	cMin	Maximum iteration	Number of population	
1	4e−5	10	4	

Below you can see the steps of the UNet-GOA:

1. Initialize the population with random solutions

2. Define the architecture of the U-Net and initialize the weights

3. Evaluate the fitness of each solution using the U-Net model

4. Set the global best solution as the one with the highest fitness

5. Repeat until convergence:

(a) Update the position of each grasshopper

(b) Evaluate the fitness of each new solution using the U-Net model

(c) Update the personal best solution for each grasshopper

(d) Update the global best solution if a better solution is found

(e) Update the step size and crowding coefficient based on the current iteration

6. Return the global best solution.

Tools

In this article, we utilized MATLAB R2018b software for both manual hyperparameter tuning of U-Net and automatic hyperparameter tuning of the proposed hybrid method (UNet-GOA). The experiments were conducted on a system running Microsoft Windows 8.1 Pro with 8GB RAM and an Intel® Core (TM) i5-7200U CPU operating at 2.50 GHz with a clock speed of 2601 MHz’s The system also featured Intel® HD Graphics 620 and a 64-bit Operating System.

Results

In this section, the results obtained from applying the U-Net architecture and the proposed UNet-GOA model on four different geometric datasets are presented side by side in both tables and plotted graphs. This visual representation facilitates easier comparison between the two methods and allows for a comprehensive analysis of their performance on the respective datasets.

The U-Net architecture, as depicted in the tables and graphics, was executed using the hyperparameter values provided in Table 2. Similarly, the grasshopper optimization algorithm in the proposed hybrid method (UNet-GOA) model was run with the hyperparameter values specified in Table 3. Moreover, for the purpose of this study, we conducted an automated search to find the optimal hyperparameters for the U-Net architecture in the proposed hybrid model, rather than manually setting them as in Table 2. The resulting best hyperparameters can be observed in Table 4.

Table 4 Optimum hyperparameter values obtained by running the proposed hybrid model (UNet-GOA) on different datasets (triangle, kite, parallelogram, and square).

	Initial learning rate	Learning rate drop factor	Learning rate drop period	Mini batch size	
Triangle	2.2652e−04	0.8420	1	3	
Kite	2.2981e−04	0.8792	1	3	
Parallelogram	2.2341e−04	0.8199	1	3	
Square	2.2147e−04	0.8541	1	3	

Figure 5 Results of image segmentation on the triangle dataset by U-Net and UNet-GOA.

In order to thoroughly evaluate the performance and effectiveness of both the U-Net architecture and the proposed hybrid method (UNet-GOA), we conducted a comprehensive series of experiments on four distinct datasets: triangle, kite, parallelogram, and square. Each dataset represents a different geometric shape and presents its own set of challenges and complexities. By subjecting the U-Net architecture and the UNet-GOA hybrid method to these diverse datasets, we were able to gather valuable insights into their capabilities and performance across a range of scenarios. The results of these experiments are extensively discussed and presented in the subsequent sections, providing a comprehensive analysis of the strengths, weaknesses, and overall suitability of the U-Net architecture and the UNet-GOA hybrid method for various dataset types.

Results of the triangle dataset

Both the U-Net and UNet-GOA models were utilized to process the triangle dataset, as shown in Fig. 5. The results, presented in Table 5 as the confusion matrix and in Table 6 as the evaluation metrics, offer a comprehensive assessment of the models’ performance. It is evident from the values in Table 5 that the proposed UNet-GOA method outperforms the U-Net model. Both models were evaluated under the same conditions with five epochs and 125 iterations per epoch, totaling 625 iterations on the triangular dataset. A comparison between the two models demonstrates significant improvements in metrics such as Accuracy (4.96% increase), Intersection over Union (7.57% increase), and BFScore (13.52% increase). These findings highlight the effectiveness and superior performance of the proposed UNet-GOA method in handling the triangle dataset.

Table 5 Results of confusion matrix in triangle dataset by U-Net and UNet-GOA.

	Triangle	Background	
	U-Net	UNet-GOA	U-Net	UNet-GOA	
Triangle	16520	18242	1908	186	
Background	1241	265	1.7103e+05	1.72e+05	

Table 6 Results of evaluation metrics in triangle dataset by U-Net and UNet-GOA.

	Accuracy	IoU	BFScore	Epoch	Iteration Per Epoch	Iteration	
U-Net	0.94463	0.91091	0.83924	5	125	625	
UNet-GOA	0.99418	0.98663	0.97442	5	125	625	
Notes.

The bold indicates best results.

Results of the Kite dataset

Figure 6 displays the use of both the U-Net and UNet-GOA models for processing the kite dataset. The outcomes are summarized in Table 7, representing the confusion matrix, and Table 8, containing the evaluation metrics, providing a comprehensive evaluation of the models’ performance. The values in Table 7 clearly indicate that the proposed UNet-GOA method outperforms the U-Net model. Both models were evaluated under the same conditions, including five epochs and 125 iterations per epoch, resulting in a total of 625 iterations on the kite dataset. A comparison between the models reveals significant improvements in metrics such as Accuracy (3.38% increase), Intersection over Union (6.83% increase), and BFScore (14.89% increase). These results emphasize the effectiveness and superior performance of the proposed UNet-GOA method in handling the kite dataset.

Results of the parallelogram dataset

Figure 7 illustrates the usage of both the U-Net and UNet-GOA models for processing the parallelogram dataset. The results obtained from these models are presented in Table 9, which showcases the confusion matrix, and Table 10, which displays the evaluation metrics. These metrics provide a comprehensive evaluation of the models’ performance and demonstrate the superiority of the proposed UNet-GOA method over the U-Net model. Both models were evaluated under the same conditions, with five epochs and 125 iterations per epoch, resulting in a total of 625 iterations on the parallelogram dataset. The comparison between the models reveals substantial improvements across metrics such as Accuracy (2.18% increase), Intersection over Union (4.77% increase), and BFScore (8.79% increase). These findings emphasize the effectiveness and superior performance of the proposed UNet-GOA method in handling the parallelogram dataset.

Figure 6 Results of image segmentation on the Kite dataset by U-Net and UNet-GOA.

Table 7 Results of confusion matrix in kite dataset by U-Net and UNet-GOA.

	Kite	Background	
	U-Net	UNet-GOA	U-Net	UNet-GOA	
Kite	23206	24625	1572	153	
Background	1788	60	1.6412e +05	1.6585e +05	
Notes.

The bold indicates best results.

Table 8 Results of evaluation metrics in the Kite dataset by U-Net and UNet-GOA.

	Accuracy	IoU	BFScore	Epoch	Iteration Per Epoch	Iteration	
U-Net	0.96289	0.92673	0.83816	5	125	625	
UNet-GOA	0.99673	0.99507	0.98704	5	125	625	
Notes.

The bold indicates best results.

Figure 7 Results of image segmentation on the parallelogram dataset by U-Net and UNet-GOA.

Results of the square dataset

Figure 8 shows the utilization of both the U-Net and UNet-GOA models for processing the square dataset. The results, presented in Table 11 (confusion matrix) and Table 12 (evaluation metrics), provide a comprehensive evaluation of the models’ performance. The proposed UNet-GOA method demonstrates superiority over the U-Net model. Both models were evaluated under identical conditions, with five epochs and 125 iterations per epoch, resulting in a total of 625 iterations on the square dataset. Notably, the UNet-GOA method outperforms the U-Net model, with significant improvements observed in metrics such as Accuracy (5.98% increase), Intersection over Union (10.76% increase), and BFScore (22.16% increase). These findings emphasize the effectiveness and superior performance of the UNet-GOA method in handling the square dataset.

Table 9 Results of confusion matrix in parallelogram dataset by U-Net and UNet-GOA.

	parallelogram	Background	
	U-Net	UNet-GOA	U-Net	UNet-GOA	
parallelogram	15277	15899	881	259	
Background	1097	196	1.7463e+05	1.7553e+05	
Notes.

The bold indicates best results.

Table 10 Results of evaluation metrics in parallelogram dataset by U-Net and UNet-GOA.

	Accuracy	IoU	BFScore	Epoch	Iteration Per Epoch	Iteration	
U-Net	0.96962	0.93708	0.88395	5	125	625	
UNet-GOA	0.99143	0.9848	0.97181	5	125	625	
Notes.

The bold indicates best results.

Figure 8 Results of image segmentation on the square dataset by U-Net and UNet-GOA.

Conclusions and Future Works

The main objective of this research was to enhance the performance of image segmentation by automating the tuning process of hyperparameters in the U-Net architecture, eliminating the need for manual configuration. In this article, a novel hybrid image segmentation method called UNet-GOA is introduced to achieve this goal. The experimental results presented in Fig. 9 clearly demonstrate the significant superiority of the proposed method compared to the U-Net model across various datasets, including triangle, kite, parallelogram, and square images, in terms of mean Accuracy, Intersection over Union, and BFScore metrics. These findings emphasize the exceptional performance of the proposed UNet-GOA method in achieving precise and robust image segmentation results.

The main aim of this study is to improve the accuracy of image segmentation by optimizing the U-Net architecture. To achieve this, we utilized the grasshopper optimization algorithm to fine-tune the hyperparameters of U-Net. However, tuning the hyperparameters of the grasshopper optimization algorithm can be challenging. In future works, we can overcome this difficulty by adopting a co-evolutionary approach and introducing the co-evolutionary version of the grasshopper optimization algorithm (CGOA). This will be the first time CGOA is used in both the literature and our future work. By incorporating CGOA, we aim to enhance the optimization process and simplify the hyperparameter tuning of the grasshopper optimization algorithm, leading to improved segmentation results. In addition, in future studies, it will be possible to compare the results by testing the proposed method on various datasets. This can further assist in evaluating the effectiveness and generalizability of the method across different data samples.

Table 11 Results of confusion matrix in square dataset by U-Net and UNet-GOA.

	square	Background	
	U-Net	UNet-GOA	U-Net	UNet-GOA	
square	21580	24116	2560	24	
Background	2424	14	1.6448e +05	1.6689e +05	
Notes.

The bold indicates best results.

Table 12 Results of evaluation metrics in square dataset by U-Net and UNet-GOA.

	Accuracy	IoU	BFScore	Epoch	Iteration Per Epoch	Iteration	
U-Net	0.93971	0.89148	0.77616	5	125	625	
UNet-GOA	0.99946	0.9991	0.99779	5	125	625	
Notes.

The bold indicates best results.

Figure 9 Comparing the results of U-Net and UNet-GOA models in different datasets.

Supplemental Information

Data S1 Dataset

Click here for additional data file.

Supplemental Information 2 Code

Click here for additional data file.

Additional Information and Declarations

Competing Interests

Author Contributions

Data Availability

The authors declare there are no competing interests.

Tohid Yousefi conceived and designed the experiments, performed the experiments, analyzed the data, performed the computation work, prepared figures and/or tables, authored or reviewed drafts of the article, and approved the final draft.

Özlem Aktaş conceived and designed the experiments, performed the experiments, analyzed the data, performed the computation work, prepared figures and/or tables, authored or reviewed drafts of the article, and approved the final draft.

The following information was supplied regarding data availability:

The Geometric Shapes Mathematics dataset, consisting of four distinct geometric images (triangle, kite, parallelogram, and square), with 1000 examples allocated for the training dataset and 200 examples for the test dataset, is available at Kaggle; Moch. Galang Rivaldo (Owner): https://www.kaggle.com/reevald/geometric-shapes-mathematics.

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
