# Peer review of "New hybrid segmentation algorithm: UNet-GOA"

_PeerJ Computer Science, doi:10.7717/peerj-cs.1499_

## Round 0.1 · original submission · Major Revisions

Dear authors,

The reviews for your manuscript are included at the bottom of this letter. We ask that you make changes to your manuscript based on those comments.

Best wishes,

Reviewer 1 ·

Basic reporting

Successful results were obtained in the segmentation study using the Grasshopper Optimization Algorithm. However, the fact that the number of images in the dataset used in the study is low and the dataset has two classes is the biggest handicap of the study. Testing the study on a similar data set will increase the quality of the article. In addition, the lack of a literature review on the subject is one of the other major shortcomings. It should be stated which parameters are optimized in the study. It is also important how many times the proposed optimization algorithm is run. Are the results presented the best or average values obtained?

Experimental design

A paragraph about the organization of the article can be added at the end of the Introduction section. The Introduction section should be reworked. There is a gap between paragraphs in the introduciton section, it is important to ensure the integrity of the subject. 2.1.3. What is the purpose of the Data Augmentation title?

Validity of the findings

3.1. Instead of using a link in the dataset header, it would be more accurate to refer to the relevant link.
There are missing sentences and many typographical errors in the study. example "Then run the Unet-GOA according to the algorithm to the optimal values and obtain a high-accuracy segmented image.Future Work", line 392.

Additional comments

It is especially important to eliminate the deficiencies on the model side. It is especially important to eliminate the deficiencies on the model side. In addition, there are many models used for segmentation in the literature. Why Unet?

Cite this review as
Anonymous Reviewer (2023) Peer Review #1 of "New hybrid segmentation algorithm: UNet-GOA (v0.1)". PeerJ Computer Science

Reviewer 2 ·

Basic reporting

- Keywords should be added in alphabetical order.
- Authors should add a paragraph into the introduction section. It may contain "The main contributions of this paper are: (i) ….. (ii) ……. and (iii) ……" to highlight the key works. By this way, authors may provide a stronger motivation clearly and explain the originality of the paper.
- In the Introduction, it is not stated what distinguishes this article from the other studies available in the literature. The gap in existing literature is not identified by arguing what is missing or inadequate in existing solutions and thus your study is required. This should be mentioned briefly in the introduction and then expanded on in the Literature Review, with in-depth analysis and citation substantiation.
- The references given in the study are not current.
- In the Introduction section, the organization of the paper is not given.
- Although there are many models used in the literature, why only UNet was used in this study.
- There are many metaheuristic algorithms and their improved versions in the literature. In this study, it was not mentioned why GOA was chosen specifically.
- GOA is a swarm-based metaheuristic algorithm. According to “A new hybrid method based on Aquila optimizer and tangent search algorithm for global optimization” and “Plant intelligence based metaheuristic optimization algorithms” papers, metaheuristic algorithms are examined in nine different classes based on the source of inspiration. These two papers should be considered for categorization and brief overview of metaheuristic algorithms.
- All variables in the equations and in the text should be written in italics. In addition, descriptions of all variables should be given in the text.
- What does the expression "minimize loss function" given as a step in Figure 4 correspond to?
- The writing format on line 310 is different. The writing format of the article should be revised.
- The expression "Then run the Unet-GOA algorithm according to the optimal values and obtain a high-accuracy segmented image.Future Work" in lines 391-393 should be corrected.
- The limitations of the proposed method do not exist.
- In conclusion, no information is given about the discussion, analysis and future studies. Conclusion section should be reorganized.

Experimental design

- It should be clearly stated what the decision variables of the GOA are and what is used as the fitness function.
- In experimental results, the hardware and software details are not presented.

Validity of the findings

- The proposed method gave successful results in the database used in the study. However, the number of samples in the data set used in the study is small. To evaluate the success of the proposed method, it should be applied to another dataset with similar characteristics.

Additional comments

See above

Cite this review as
Anonymous Reviewer (2023) Peer Review #2 of "New hybrid segmentation algorithm: UNet-GOA (v0.1)". PeerJ Computer Science

---

## Round 0.2 · Minor Revisions

Dear author,

Your article has a few remaining issues. We encourage you to address the concerns and criticisms of the reviewer and resubmit your article once you have updated it accordingly.

Reviewer 1 has requested that you review one specific reference, but I do not see its relevance to your work. You may add it if you believe it is especially relevant, although its inclusion (or exclusion) will not influence my decision.

Best wishes,

Reviewer 1 ·

Basic reporting

Congratulations to the researchers for this comprehensive revision. Researchers have corrected major deficiencies in the revision. But after some minor changes, I think the work can be accepted. Heading 3.1.2 should be replaced with a more appropriate one. In Title 3.3, the usage areas of Meta-Heuristic Algorithms are mentioned. Meta-Heuristic algorithms can be used for different purposes. I recommend you review the article "Multi-feature fusion and improved BO and IGWO metaheuristics based models for automatically diagnosing the sleep disorders from sleep sounds". It is quite difficult to read the values in some confusion matrices. It is important to update low-resolution figures. The conclusion is the last part of the study. I think it is not correct to cite references in this section.

Experimental design

.

Validity of the findings

.

Additional comments

.

Cite this review as
Anonymous Reviewer (2023) Peer Review #1 of "New hybrid segmentation algorithm: UNet-GOA (v0.2)". PeerJ Computer Science

---

## Round 0.3 · accepted · Accept

Dear authors,

Thank you for clearly addressing all of the reviewers' comments. Your article is accepted for publication after your second revision.

Best wishes,